# Optimization of Resveratrol Used as a Scaffold to Design Histone Deacetylase (HDAC-1 and HDAC-2) Inhibitors

**DOI:** 10.3390/ph15101260

**Published:** 2022-10-13

**Authors:** Beatriz Silva Urias, Aline Renata Pavan, Gabriela Ribeiro Albuquerque, Igor Muccilo Prokopczyk, Tânia Mara Ferreira Alves, Thais Regina Ferreira de Melo, Geraldo Rodrigues Sartori, João Hermínio Martins da Silva, Chung Man Chin, Jean Leandro Dos Santos

**Affiliations:** 1School of Pharmaceutical Sciences, São Paulo State University (UNESP), Araraquara 14800-903, SP, Brazil; 2Institute of Chemistry, São Paulo State University (UNESP), Araraquara 14800-060, SP, Brazil; 3Laboratory of Structural and Functional Biology Applied to Biopharmaceuticals, Oswaldo Cruz Foundation (Fiocruz), Eusébio 61773-270, CE, Brazil; 4Postgraduate Program in Computational and Systems Biology, Oswaldo Cruz Foundation (Fiocruz), Rio de Janeiro 21040-222, RJ, Brazil

**Keywords:** histone deacetylase, resveratrol, enzymatic inhibition, gene regulation, new drugs

## Abstract

Histone deacetylases (HDAC) are epigenetic enzymes responsible for repressing gene expression through the deacetylation of histone lysine residues. Therefore, inhibition of HDACs has become an interesting approach for the treatment of several diseases, including cancer, hematology, neurodegenerative, immune diseases, bacterial infections, and more. Resveratrol (RVT) has pleiotropic effects, including pan-inhibition of HDAC isoforms; however, its ability to interfere with membranes requires additional optimization to eliminate nonspecific and off-target effects. Thus, to explore RVT as a scaffold, we designed a series of novel HDAC-1 and -2 inhibitors containing the 2-aminobenzamide subunit. Using molecular modeling, all compounds, except unsaturated compounds (**4**) and (**7**), exhibited a similar mode of interaction at the active sites of HDAC 1 and 2. The docking score values obtained from the study ranged from −12.780 to −10.967 Kcal/mol. All compounds were synthesized, with overall yields ranging from 33% to 67.3%. In an initial screening, compounds (**4**), (**5**), (**7**), and (**20**)–(**26**), showed enzymatic inhibitory effects ranging from 1 to 96% and 6 to 93% against HDAC-1 and HDAC-2, respectively. Compound (**5**), the most promising HDAC inhibitor in this series, was selected for IC_50_ assays, resulting in IC_50_ values of 0.44 µM and 0.37 µM against HDAC-1 and HDAC-2, respectively. In a panel of selectivity against HDACs 3–11, compound (**5**) presented selectivity towards Class I, mainly HDAC-1, 2, and 3. All compounds exhibited suitable physicochemical and ADMET properties as determined using in silico simulations. In conclusion, the optimization of the RVT structure allows the design of selective HDAC inhibitors, mainly targeting HDAC-1 and HDAC-2 isoforms.

## 1. Introduction

Epigenetic modifications are involved in chromatin remodeling and in altering gene expression. Acetylation of lysine residues is one of the most well-described post-translational modifications in epigenetic mechanisms. The two main enzymes involved in this mechanism are histone acetyltransferase (HATs) and histone deacetylase (HDACs). HAT acetylates the lysine residues in the histone tail to neutralize the amino acid charge and eliminate its interaction with the DNA, resulting in chromatin relaxation and facilitating access to the genetic material by RNA polymerase, leading to gene expression. Contrarily, histone deacetylases (HDAC) are a family of enzymes responsible for removing acetyl groups, restoring lysine charge, and interacting with DNA, hindering the entry of RNA polymerase resulting in gene silencing [1,2,3,4] (Figure 1).

Eighteen HDAC isoforms have been described and classified into four classes (class I–IV), according to their homology and localization in cells. HDAC comprises the following isoforms: class I (HDAC-1, 2, 3, and 8), class IIa (HDAC-4, 5, 7, and 9), class IIb (HDAC-6 and 10), class III (SIRT1–7), and class IV (HDAC-11). Regarding their mechanism of action, classes I, II, and IV are zinc-dependent enzymes, while class III includes NAD+-dependent enzymes, also called sirtuins (SIRTs) [5,6]. HDACs have been described as promising targets in several diseases, including cancer (solid and hematological malignancies) [7], neurological, autoimmune, inflammatory, metabolic disorders, and others. Despite their potential to treat diseases, few FDA-approved drugs have reached the market. Among the drugs licensed for use as HDAC inhibitors are: vorinostat (Zolinza) [8]; romidepsin (Istodax) [9]; belinostat (Beleodaq) for cutaneous T-cell lymphoma [10]; and panobinostat (Farydak) [11] for the treatment of multiple myeloma. China’s National Medical Products Administration has also approved tucidinostat (Epidaza) [12] for the treatment of peripheral T-cell lymphoma [13,14] (Figure 2). The reasons for their limited therapeutic use are severe hematological (thrombocytopenia, neutropenia) and cardiac (ventricular arrhythmia) adverse effects related to their lack of selectivity for inhibiting the other distinct isoforms. Thus, approaches aimed at designing selective inhibitors are widely desirable to address off-target and adverse effects of HDAC inhibitors [15]. Structural requirements for the design of selective HDAC inhibitors have been described elsewhere [16,17,18,19,20,21]. In general, the scaffold of an HDAC inhibitor is constituted by a zinc-binding group, a hydrophobic linker, and an exposure “cap” (Figure 2).

Resveratrol (RVT) (trans-3,5,4′-trihydroxystilbene) (Figure 3) is a natural product with pleiotropic effects, acting on several diseases such as cancer, cardiovascular, metabolic, neurodegenerative, and inflammation [22,23,24,25]. Experiments using nuclear extract of HeLa cells revealed that RVT inhibited all HDAC isoforms at 100 µM but was most active against HDAC-1 and HDAC-10 followed by HDAC-4 and HDAC-9, with 40% and 50% inhibitory values, respectively [26]. In addition to HDAC classes I, II, and IV, RVT also activates SIRT [27]. Dose-dependent experiments showed that RVT increased the deacetylation rate of SIRT1 [27]. The activation of SIRT1 has been reported to be one of the modes of action of RVT for metabolic disorders, cardioprotection, cancer, and neuroprotection [28]. Although RVT is an interesting prototype as an HDAC inhibitor, its poor potency, lack of selectivity, and nonspecific effects due to its bilayer membrane-perturbation effects demand additional efforts to optimize its HDAC inhibition [29,30]. Previously, we have described the potential of RVT analogs as inducers of fetal hemoglobin in sickle cell disease by inhibiting HDACs without membrane perturbation effects [30].

Therefore, in a continuing effort to design RVT derivatives with HDAC inhibition, we described herein the synthesis and evaluation of novel analogs containing the 2-aminobenzamide subunit. This subunit in tucidinostat (selective class I) acts as a zinc-binding group. We carried out molecular modeling studies to improve their selectivity against HDAC class I and explored chemical modifications in the linker and “cap” regions, to further comprehend their rigidity and polarity (Figure 3).

## 2. Results and Discussion

### 2.1. Molecular Docking Studies and In Silico Prediction of ADME Properties

The structural basis for RVT optimization was assisted by molecular docking, which suggested the poses and modes of interaction between HDAC-1 and HDAC-2. For RVT, the DS values were found to be −5.565 and −6.453 against HDAC-1 and HDAC-2, respectively. Both non-ionized and ionized phenol groups were considered in this study. RVT fits into the binding pocket, but we did not observe any interactions with zinc, as previously reported [26]. Figure 4 shows that in the HDAC-1 active site, RVT can interact with residues ARP176 through hydrogen bonding and with HIS141 and HIE178 via π- π interactions. In the HDAC-2 binding pocket (Figure 5), RVT interacts with GLY143 and PHE155 by hydrogen bonding and π-π, respectively. The limited number for RVT explains its poor DS value compared to the optimized compounds described here. Appendix A show the pose of RVT in the active site of HDAC-1 and HDAC-2, respectively.

Computational studies guided the type (H, OH, NH_2_, and CONH_2_) and pattern (ortho and para) of substitution for all compounds, and it was carried out to provide additional interactions with the enzymes. The insertion of 2-aminobenzamide as a zinc chelating subunit allows the orientation of this group toward the metal at the active site of the enzyme. Table 1 shows the DS values for HDAC-1 and HDAC-2, ranging from −4.462 to −10.763 Kcal/mol for HDAC-1 and −10.971 to −12.780 Kcal/mol for HDAC-2. For HDAC-2, redocking procedures were performed since the co-crystallized ligand (20Y) was already described. Notably, the root mean square deviation (RMSD) of 20Y for HDAC-2 was 0.46 Å.

Except for compounds (**4**) and (**7**), the poses of the compounds revealed that, in addition to the 2-aminobenzamide subunit occupying the active site of HDAC-2, there were interactions involving hydrogen bonds with HIS145 and/or HIS146 and GLY154. The rigidity conferred by the unsaturation of compounds (**4**) and (**7**) did not allow an appropriate fit against HDAC-1 and HDAC-2. Structural differences between both isoforms, such as a wider entrance in the active site for HDAC-2 [16], could explain the change in the pattern of the poses of unsaturated compounds.

Compound (**5**), a saturated analog of compound (**4**), presented an improved DS value with the 2-aminobenzamide moiety occupying the active site for both HDAC-1 and HDAC-2 (Figure 6). The flexibility conferred by the C-C sp3 bond of (**5**) enables its rotation and improves its fit at the active site of both enzymes. Moreover, an additional hydrogen bond was found involving the amino group (NH_2_) in the cap and LEU276 or GLU208 in comparison with compound (**4**) (Appendix A).

For all compounds containing a flexible linker (**5**; **20**–**26**), no significant differences in DS values were found when comparing the presence of the following atoms: carbon (**5**), nitrogen (**21**, **24**), and oxygen (**20**, **22**–**26**) at this linker. However, compounds presenting polar subunits in the cap, such as (**5**), (**22**), (**23**), and (**24**), presented an additional hydrogen bond with an external LEU276, which was not seen in other compounds (Appendix A). For some compounds, such as (**20**), (**21**), and (**24**), additional π- π interaction with TYR209 residue was observed, but with no significant changes in DS values (Appendix A).

Protection of the amine group of compound (**23**), with bulky phthalimide groups (**25**, **26**), did not demonstrate any additional interactions with external amino acids. In addition, this group was unfavorably exposed (Appendix A).

In order to evaluate the reliability of proposed poses by docking, we simulated the compound (**5**) complexed with HDAC-1 and HDAC-2 and also the compound (**4**) in complex with HDAC-2. In all simulations, the protein remained quite stable, with RMSD below 2 Å (Appendix A). Due to the high rigidity, compound (**4**) showed oscillating RMSD below 3 Å during all five replicates (Appendix A). Despite unfavorable exposure of the 4-aminophenol moiety to the solvent, the aminobenzamide group strongly interact with Zn^2+^ and residues HIS145, HIS146 and GLY154 and remain in the active site during the 50 ns simulations, despite being a weaker inhibitor than (**5**). Specifically, the ligands interact with HIS145 during more than 85% of frames in all simulations, and during around 40–50% of frames with HIS146 and GLY154 (Appendix A)

On the other hand, the compound (**5**) showed higher value of RMSD when in complex with both HDAC-1 or HDAC-2, without difference between the proteins (Appendix A). However, despite the high value it is observed very stable plateau at 2 and 4 Å along the simulations. This is easily explained by two possible positioning of 4-aminophenyl moiety in the surrounding of the protein cavity (Figure 7). The first one is similar to the proposed by the docking and group seems to do not perform any stable interaction with protein (Figure 7, green line). The latter is characterized by the T-stacking of 4-aminophenyl with PHE155 (Figure 7, green stick). This interaction is already known and is present in many of the existing crystallographic structures of HDAC-2. However, differently to those available in PDB, the compound (**5**) also presents a second phenyl ring that perform a π-stacking interaction with the PHE155 and thus define a triplet cluster of aromatic rings (Figure 7, green stick). It is needed to reinforce this interaction but is still not enough to stabilize the 4-aminophenyl near the PHE155, since we observed few flips between the conformations (Appendix A).

In silico predictions of ADME properties were obtained by the Swiss ADME software (Appendix A). Drug-like, ADMET, and physicochemical properties were characterized to identify those compounds that are most promising for further evaluation. For all derivatives, cLogP values ranged from 2.12–3.21, being more lipophilic than RVT (cLogP 1.71). Tucidinostat exhibited a cLogP value of 2.51. RVT, tucidinostat, and compounds (**23**)–(**24**) were considered soluble, while the others were moderately soluble. All compounds are predicted to have high intestinal absorption, except for compound (**26**), which is predicted to have low intestinal absorption. CYP2C9 inhibition is predicted in all compounds. Regarding the Lipinski rule, except for compound (**26**), all compounds presented a molecular weight lower than 500 daltons; a cLogP value lower than 5, less than 5 hydrogen bond donors, and less than 10 hydrogen bond acceptors. Therefore, all synthesized compounds comply with Lipinski’s rule of five.

### 2.2. Chemistry

The synthetic routes to obtain compounds (**4**)–(**5**), (**7**), (**20**)–(**24**), and (**25**) and (**26**) are shown in Figure 8, Figure 9 and Figure 10. For compound (**5**), initially, 4-bromobenzoic acid (1) was treated with oxalyl chloride in dichloromethane (DCM) and then added to *o*-phenylenediamine in a basic medium to provide an intermediate (**2**) at a yield of 65%. As a next step, protection of the amine group was carried out by introducing the protector group Boc, followed by the Heck [31] reaction with 4-vinylaniline and palladium acetate as catalysts to produce compound (**4**) at a yield of 33%. The final reaction was catalyst hydrogenation with palladium on carbon (Pd/C) and hydrogen gas (H_2_) to obtain compound (**5**) at a yield of 90%. Compound (**8**) has been submitted for the same reactions but in a different sequence. Initially, 4-bromobenzoic acid (**1**) was coupled to styrene through the Heck reaction to provide an intermediate (**6**) at a yield of 60%. The final step was the treatment of compound (**6**) with oxalyl chloride and then the coupling with o-phenylenediamine, providing compound (**7**) with a yield of 6% (Figure 8).

For compounds (**20**)–(**24**), 4-bromomethyl benzoic acid (**8**) was protected by esterification with methanol and sulfuric acid [32] and then reacted through SN2 [33] with phenol, aniline, 4-aminophenol, 4-hydroxyphenol, or salicylamide, to provide intermediates (**10**)–(**14**) at yields ranging from 50 to 82%. The final step was the ester hydrolysis and coupling reaction with o-phenylenediamine through coupling agent 1,1-carbonyldiimidazole (CDI), providing compounds (**20**)–(**24**) with yields ranging from 10 to 40% (Figure 9).

Compound (**25**) was treated with coupling agents 1-ethyl-3-(3-dimethylaminopropyl)carbodiimide (EDC) and 4-dimethylaminopyridine (DMAP) and coupled with phthalic or 3-nitrophthalic anhydride to obtain compounds (**25**) and (**26**) at yields ranging from 15 to 25% (Figure 10).

### 2.3. Enzymatic Evaluation

Enzymatic assays to evaluate HDAC inhibition were performed, and the design involved three steps: (a) Preliminary screening to identify the most promising compounds; (b) Evaluation by a panel of HDAC isoforms to investigate selectivity; and (c) Determination of IC_50_ values. Firstly, compounds (**4**–**7**; **15**–**26**) were evaluated at 10 µM against HDAC-1 and HDAC-2 (Table 2). Intermediates (**6**; **15**–**19**) and finally (**4**, **5**, **7**, **20**–**24**) compounds evaluated at 10 µM showed inhibitory activity against HDAC-1 ranging from 0–96%, while for HDAC-2 those values ranged from 0–93%. All intermediates (**6**; **15**–**19**) exhibited no-effect or very weak inhibitory effect against both HDAC-1 and HDAC-2, with compound (**15**) being the most active among the series, inhibiting HDAC-1 by 16% whereas HDAC-2 by 11%. These results were expected due to the lack of the zinc-binding subunit in these structures.

Of the final compounds (**4**, **5**, **7**, **20**–**26**), the most active ones were (**5**, **20**, **21**, **22**, and **24**) by inhibiting up to 96% of HDAC-1 and 93% of HDAC-2 enzymatic activity. The conformational restriction provided by the double bond does not favor the inhibitory effect, as previously observed in the docking studies. Compounds (**4**) and (**7**) presented inhibitory values of 45% and 42% against HDAC-1 and 28% and 32% against HDAC 2, which are up to three times lower in comparison with compound (**5**). For compound (**5**) the inhibitory values against HDAC-1 and HDAC-2 were 92% and 93%, respectively.

Compounds (**20**)–(**24**) exhibited inhibition values against HDAC-1 ranging from 72% to 96% and against HDAC 2 from 52% to 93%. These compounds contain a greater polar link than (**5**), suggesting that such an effect is not desirable for HDAC inhibition. Compounds (**20**) and (**21**) have been already published in the literature as HDAC inhibitors [33,34]. Compound (**20**) had its HDAC inhibition activity evaluated only in the enzymes from the HeLa nuclear cells, while compound (**21**) presented an IC_50_ of 1 ± 0.1 µM against HDAC-1 and 1.4 ± 0.05 µM against HDAC-2. The comparison between (**20**), (**22**), and (**23**), showed that the amino group in the position para of the ‘cap’ has an improved effect compared to the H (**20**) and hydroxy (**22**) groups. Interestingly, the presence of bulky protecting amine groups in the ‘cap’ as represented by compounds (**25**) and (**26**) led to a reduction in the effect since their inhibition values were 14% and 1% for HDAC-1 and 21 and 6% for HDAC-2, respectively.

Based on the results, compound (**5**) was selected for further assays to evaluate its selectivity against the HDACs (4–11) and the determination of IC_50_ values (Table 3). At a concentration of 10 µM, compound (**5**) inhibited HDAC-3 and HDAC-10 by 82% and 60%, respectively. For other HDAC isoforms, values were found to be equal to or lower than 15%. The value of IC_50_ for HDAC-1 and HDAC-2 was 0.44 µM and 0.37 µM, respectively.

The results seen in the molecular docking studies contributed to comprehend and predict the trends in the activity of the compounds. As shown, the docking score and the pose predicted for compound (**4**) and (**5**) have already demonstrated a difference in the interaction pattern and in the mode of binding between these molecules due to the conformational rigidity changes, with (**5**) being highlighted as the most promisor between both compounds. A similar result was seen in the enzymatic evaluation, with compound (**5**) being at least 2-fold more active than compound (**4**) against both HDAC-1 and HDAC-2. Regarding compounds (**20**)–(**24**), the docking study showed a similar mode of binding among them and, besides a few differences in the interactions with the amino acid residues. DS values did not change much when the data from HDAC-2 was analyzed. A similar trend was seen in the experimental results, with the inhibition values ranging from 72 to 96% against HDAC-1 and from 52 to 93% against HDAC-2. Compounds (**25**) and (**26**) were developed with an additional protection in the cap region, by the addition of the phthalimide subunit. The in silico study demonstrated that the cap region of these molecules was exposed to the solvents, and was not able to perform any additional interaction with the active site. The enzymatic evaluation demonstrated that both compounds were not able to inhibit HDAC-1 and HDAC-2 efficaciously. Although compounds (**5**), (**21**), and (**22**), were the most active inhibitors for HDAC-1 and HDAC-2, we selected compound (**5**) for the IC_50_ and selectivity studies. The reason for that was due to favorable pharmacodynamic and ADME profiles. It is well established in the literature the importance of the hydrophobic character of the linker. As described [16], the 11 Å channel in the active site of HDAC-1 and HDAC-2 presents hydrophobic amino acid residues, which was taken into account in the decision of compound (**5**) compared to (**21**) and (**22**). The presence of arilamine in addition, the intermediate clogP values (logP < 3), and good water solubility for (**5**), motivated the continuity of the studies with this compound. We found no compound (**5**) sub micromolar IC_50_ values against HDAC-1 and HDAC-2, and an interesting selectivity towards HDAC Class I.

## 3. Materials and Methods

### 3.1. Computational Methods

#### 3.1.1. Molecular Docking Studies

In silico studies were performed on Schrodinger (2019–4) Maestro v12.2 molecular modeling environment and MarvinSketch on a computer containing an Intel Core I7–4790 processor with 16 Gb memory and the Nvidia GeForce GTX 980 graphic processor. All the 2D structures were drawn using MarvinSketch. The 3D structures were generated by the LigPrep procedure. Tautomers and stereoisomers were assessed. For this study, the targets used included HDAC-1 (PDB ID: 4BKX; resolution: 3.00 Å) and HDAC-2 (PDB ID: 4LY1; resolution: 1.57 Å), which were retrieved from Protein Data Bank. Protein preparations were carried out on the Protein Preparation Wizard, following the steps: (i) Removal of water molecules; (ii) Adding hydrogen atoms; (iii) Filling in incomplete side chains; (iv) Energy minimization using OPLS3 force field. It was used in Epik to generate the protonation state at pH 7.0 ± 2.0 and OPLS03 as a field force [35]. The validation of the molecular docking method for HDAC-2 was performed by redocking of the crystalized compound (20Y) and the calculation of the Root Mean Deviation Square (RMSD). The minimization steps were repeated until the converge threshold RMSD was equal to 0.15 Å. The grid generation was prepared with volume appropriated to cover all investigational active sites. The interaction box (measuring 15 Å × 15 Å × 15 Å) was centered on the Zn+2 atom. All docking calculations were performed using Glide in extra-precision mode (XP). We performed re-docking to check for all docking simulations, 20 poses were generated, and post-docking minimization was performed. All these parameters were kept as default. The compounds were scored by GlideScore [36]. The poses were analyzed by PoseView and UCSF Chimera.

#### 3.1.2. Molecular Dynamics Simulations

The best ranked pose of complexes HDAC-1-(5), HDAC-2-(5), and HDAC-2-(4), generated by docking, as well as the free HDAC-2 protein, were selected as input structure for four batches of molecular dynamics simulations (MD). N and C-terminal were capped with ACE and NME residues, respectively. The three structural ions (Zn^2+^ and two K^+^ for HDAC-1, and Zn^2+^, Ca^+^ and Na^+^ for HDAC-2) were retrieved, respectively, from PDB IDs: 4BKX and 4LY1. All hydrogen atoms were removed to avoid nomenclature mismatch. Except for HIS286, considered as HIP, no protonation change was identified using the PDB2PQR software [37,38] at pH 7.0 and thus all other residues were kept at their default state. The histidine tautomers were defined as HID to residues 44, 73, 145, 146, 172, and 349, and as HIE to residues 33, 38, 62, 183, 184, and 204 (HDAC-2 numbering), to optimize the protein hydrogen-bond network. The RESP charge of ligands (4) and (5) was obtained using the default parameters of R.E.D server [39,40], except for CHR_TYP = RESP-X1. Antechamber and tLeap features of AmberTools21 were applied to generate a suitable ligand library in gaff2 force field for complex preparation. The previously prepared protein and parametrized ligand were parametrized using gaff2 and ff19SB [41] forcefields and solvated by a truncated octahedral box of OPC water, centered at the center of mass of protein and with edges located at least 12 Å from any protein atom. Na^+^ and Cl^-^ for neutralization and salt concentration of 120 mM. Finally, using parmed, we implemented the hydrogen mass repartitioning (HMR) [42]. Four minimizing steps was performed. Firstly, 2000 steps of minimization were executed in CPU version of pmemd implemented in Amber20 [43,44], restraining the protein, structural ions, and ligand, with force of 500 kcal.mol^−1^. After this, all minimization/simulation protocols were executed in the GPU version of pmemd. We performed an extra 7000 steps of minimization of solvent, followed by another 7000 steps restraining only the protein residues. The first half of the steps in each execution was carried out using the steepest descent method, which was switched to conjugate gradient. The last minimization protocol consisted in a minimization without restraint until convergence. Each system was linearly heated to 310K at NVT ensemble Berendsen thermostat non-solvent constraint with force of 10 kcal.mol**^−^**^1^ during 200 ps, followed by density equilibration during 500 ps at NPT ensemble, using the Langevin thermostat and Monte Carlo barostat, with collision frequency and pressure relaxation time of 2 ps**^−^**^1^ and 1 ps, respectively, also constraining non-solvent atoms with a force of 10 kcal.mol**^−^**^1^. We thus equilibrated the system during 5 ns without constraints, using the same parameters of density equilibration. After equilibration, each complex was simulated during 50 ns. All protocol was replicated five times for each complex, changing only the random seed. A timestep of 4 fs was used, enabled by the HMR and SHAKE algorithm, and the PME long-range interaction cutoff was defined as 10 Å. Simulation analysis were carried out using the cpptraj software.

#### 3.1.3. In Silico Prediction of ADME Properties

In silico ADME properties of the compounds (**4**), (**5**), (**7**), (**20**)–(**26**), were determined using the Swiss ADME software.

### 3.2. Chemistry

#### 3.2.1. General Chemical Aspects

Solvents and reagents were purchased from commercial suppliers, and for reactions, all solvents were dried before use. These reactions were monitored using thin-layer chromatography (TLC), precoated with silica gel 60 (HF-254; Sigma-Aldrich, St. Loius, MA, USA) to a thickness of 0.25 mm. The plates were exposed to UV light (254 nm) and, when necessary, treated with ninhydrin to detect primary amines. All compounds were purified on a chromatography column with silica gel (60 Å pore size, 35–75 µM particle size) using appropriate mobile phases as described for each compound. The purity of all compounds was characterized by HPLC using a Shimadzu LC-10AD chromatograph equipped with a model SPD-10A UV–vis detector (Shimadzu, Kyoto, Japan). For this study, all compounds exhibited purity greater than 98.5%. Melting points (MPs) were determined in open capillary tubes using an electrothermal melting point apparatus (SMP3; Bibby Stuart Scientific, Stone, UK). Nuclear magnetic resonance (NMR) spectra for 1H and 13C of all compounds were obtained on a Bruker DRX-600-megahertz (MHz) NMR spectrometer (Billerica, MA, USA) using deuterated solvents for sample preparation. Chemical shifts were expressed in parts per million (ppm) relative to tetramethylsilane. The coupling constants were reported in hertz (Hz), and the signal multiplicities were reported as singlet (s), doublet (d), doublet of doublets (dd), doublet of doublets of doublets (ddd), triplet (t), and multiplet (m). Mass spectra were acquired using the LC-DAD-ESI system from Shimadzu HPLC (CBM20A) (Shimadzu, Kyoto, Japan), LC-20AD quaternary pump, SPD-M20A detector, SIL-20A autosampler, and CTO-20A column compartment, coupled to a Bruker Ion Trap ESI source (Amazon SL). Mass analysis was performed in positive mode and m/z scanned 50–1000 using the following parameters: source voltage of 4.5 kV, 9.00 L/min sheath gas, 40 psi nebulizer, and dry temperature of 300 °C.

#### 3.2.2. General Procedure for the Synthesis of Compound (**2**)

In the first step, 2.5 mmol of 4-bromobenzoic acid (**1**) was kept under stirring at room temperature with an excess of oxalyl chloride (4 mmol) and 2 drops of dimethylformamide (DMF) in anhydrous DCM for 1 h. Subsequently, the solvent was removed under a vacuum and stored. In the second step, two equivalents of o-phenylenediamine (5 mmol) were added to anhydrous tetrahydrofuran (THF) and 4-methylmorpholine (10 mmol) and kept under stirring at 0 °C for 30 min. Later, the stored reaction medium was resuspended with THF and added dropwise to the amine solution. The reaction was kept under stirring conditions at room temperature (RT) for 12 h. The solvent was removed under vacuum, resuspended in 100 mL of ethyl acetate, and washed with distilled water (3 × 50 mL). The organic layer was dried over sodium sulfate, and the solvent was removed under a vacuum. A column chromatography (flash silica; eluent: ethyl acetate: hexane, 5:5 (*v/v*)) purification of the crude product provided compound (2). White powder; yield: 65%; ^1^H NMR (300 MHz, *DMSO_d6_*, δ ppm) δ: 9.80 (1H, s, NH), 7.94 (d, *J* = 8.4 Hz, 2H), 7.72 (d, *J* = 8.4 Hz, 2H), 7.15 (d, *J* = 7.6 Hz, 1H), 6.97 (t, *J* = 7.6 Hz, 1H), 6.77 (dd, *J* = 8.0 Hz, 1H), 6.59 (t, *J* = 7.3 Hz, 1H), 4.94 (s, NH_2_, 2H). ^13^C NMR (75MHz, *DMSO_d6_*, δ ppm) δ: 164.4, 143.3, 133.8, 131.2, 130.0, 126.8, 126.6, 125.1, 123.0, 116.1, 116.0.

#### 3.2.3. General Procedure for the Synthesis of Compound (**4**)

A mixture of compound (**2**) (1.7 mmol), di-tert-butyl dicarbonate (2.55 mmol), and THF were stirred at RT for 12 h. The solvent was removed under vacuum, resuspended in 100 mL of ethyl acetate, and washed with distilled water (3 × 50 mL). The organic layer was dried over sodium sulfate, and the solvent was removed under vacuum to obtain compound (**3**) as a brown oil. In a separate flask, one equivalent of compound (**3**) (1.7 mmol) was added to 3 mL of triethanolamine, 4-vinylaniline (1.7 mmol), palladium acetate II (5 mol%) and stirred at 120 °C for 24 h. The reaction medium was diluted with 100 mL of ethyl acetate and washed with distilled water (5 × 50 mL). The organic layer was dried over sodium sulfate, and the solvent was removed under a vacuum. The crude product was purified by column chromatography (flash silica; eluent: ethyl acetate: hexane, 3:7 (*v/v*)), providing the stilbene at a yield of 33%. The stilbene was treated with trifluoroacetic acid to provide a compound (**4**). Yellow powder; yield, 33%; ^1^H NMR (300 MHz, *DMSO_d6_*, δ ppm) δ: 9.64 (s, NH), 7.95 (d, *J* = 8.1 Hz, 2H), 7.62 (d, *J* = 8.1 Hz, 2H), 7.33 (d, *J* = 8.4 Hz, 2H), 7.23 (d, *J* = 16.3 Hz, 1H), 7.17 (d, *J* = 7.6 Hz, 1H), 6.99–6.96 (m, H-12, 2H), 6.79 (t, *J* = 7.6 Hz, 1H), 6.62–6.57 (m, H-15, 3H), 5.3 (d, NH_2_). ^13^C NMR (75MHz, *DMSO_d6_*, δ ppm) δ: 165.0, 149.2, 143.1, 141.1, 132.0, 131.1, 128.2, 128.1, 126.7, 125.4, 124.4, 123.5, 121.7, 116.3, 116.2; MS/ESI m/z calculated for C_21_H_19_N_3_O: 329.15, found: [M+H]^+^ 330.19.

#### 3.2.4. General Procedure for the Synthesis of Compound (**5**)

The medium containing compound (**4**) (1.5 mmol) was added to 10 mL of anhydrous acetone and Pd/C (10 mol%). H_2_ was influxed, and the reaction was stirred at RT for 6 h. The reactional medium was filtered through Celite**^®^**, washed with acetone, and the solvent was then removed under vacuum. Then, 100 mL of ethyl acetate was added and washed with distilled water thrice (3 × 50 mL). The organic layer was dried over sodium sulfate, and the solvent was removed under vacuum to obtain the compound (**5**). White powder; yield, 90%. ^1^H NMR 300 MHz, *DMSO_d6_*, δ ppm) δ: 9.59 (1H, s, NH), 7.88 (2H, d, H-9), 7.32 (2H, d, H-10), 7.17 (1H, d, H-3), 6.97 (1H, t, H-5), 6.88 (2H, d, H-15), 6.78 (1H, d, H-6), 6.60 (1H, t, H-4), 6.48 (2H, d, H-16), 4.89 (4H, s, NH_2_), 2.89 (2H, t, H-12), 2.75 (2H, t, H-13); ^13^C NMR (75MHz, *DMSO_d6_*, δ ppm) δ: 166.7, 146.0, 144.1, 142.0, 135.3, 129.7, 129.5, 129.2 127.2, 125.3, 124.7, 118.1, 116.6, 116.2, 35.6; MS/ESI m/z calculated for C_21_H_21_N_3_O: 331.16, found: [M+H] ^+^ 332.17.

#### 3.2.5. General Procedure for the Synthesis of Compounds (**6**)

One equivalent of 4-bromobenzoic acid (**1**) (2.98 mmol), styrene (2.98 mmol), palladium acetate II (5 mol%), triethanolamine (3 mmol), and 8 mL of anhydrous toluene, were stirred at 120 °C for 24 h. The solvent was removed under vacuum, resuspended in 100 mL of ethyl acetate, filtered through Celite**^®^**, and washed with distilled water (3 × 50 mL). The organic layer was dried over sodium sulfate, and the solvent was removed under a vacuum. The crude product was purified by column chromatography (flash silica; eluent: ethyl acetate: hexane, 5:5 (*v/v*)) to provide the compound (**6**). White powder; yield 60%; ^1^H NMR (300 MHz, *DMSO_d6_*, δ ppm) δ: 8.13 (d, *J* = 8.0 Hz, 2H), 7.70 (d, *J* = 8.0 Hz, 2H), 7.42 (dd, *J* = 7.7 Hz, 2H), 7.31 (t, *J* = 7.7 Hz, 2H), 7.25 (d, J = 16 Hz, 1H), 7.20 (d, J = 16 Hz, 1H), 7.16 (t, 1H). ^13^C NMR (75MHz, *DMSO_d6_*, δ ppm) δ: 167.8, 137.3, 135.6, 133.4, 130.1, 129.7, 128.8, 128.1, 127.4, 126.0

#### 3.2.6. General Procedure for the Synthesis of Compound (**7**)

One equivalent of compound (**6**) (1.10 mmol) was treated with an excess of oxalyl chloride (1.55 mmol), two drops of DMF, and 2 mL of toluene. The reaction was stirred at 70 °C for 2 h. The solvent was removed under a vacuum and stored. Two equivalents of o-phenylenediamine (2.2 mmol) were added to 250 µL of triethylamine (1.87 mmol), and 10 mL of DCM at 0 °C and the reaction was stirred for 1 h. Subsequently, the stored reaction medium was resuspended with dichloromethane and added dropwise to the amine solution. The reaction was stirred at room temperature for 24 h. The solvent was removed under vacuum, resuspended in 100 mL of ethyl acetate, and washed with sodium bicarbonate saturated solution (3 × 50 mL) and distilled water (2 × 50 mL). The organic layer was dried over sodium sulfate, and the solvent was removed under a vacuum. The crude product was purified by column chromatography (flash silica; eluent: ethyl acetate: hexane, 5:5 (*v/v*)) to provide a compound (**7**). Yellow powder; yield, 6%; mp 177–179 °C. ^1^H NMR (300 MHz, *DMSO_d6_*, δ ppm) δ: 9.17 (s, NH), 8.0 (d, 2H), 7.77 (d, 1), 7.72 (d, 2H), 7.65 (d, 1H), 7.40 (d, 2H), 7.30 (m, 2H) 7.0 (d, 1H), 6.86 (t, 1H), 6.67 (d, 1H), 4.64 (s, NH_2_). ^13^C NMR (75 MHz, *DMSO_d6_*, δ ppm) δ: 165.1, 143.7, 142.0, 137.9, 134.9, 132.3, 130.5, 129.6, 128.8, 127.6, 127.5, 127.4, 127.2, 126.2, 124.9, 118.1; MS/ESI m/z calculated for C_21_H_18_N_2_O: 314.14, found: [M+H]^+^ 315.16.

#### 3.2.7. General Procedure for the Synthesis of Compounds (**10**)–(**13**)

In the first step, 4-bromomethylbenzoic acid (**8**) was treated with 10 mL of methanol and 1 mL of sulfuric acid, and the reaction was stirred for 12 h at RT. The reaction medium was placed on an ice bath, and the product (**9**) was precipitated followed by filtration to yield 94.3% as a white powder. In the next step, one equivalent of compound (**9**) (1.31 mmol), phenol (1.31 mmol), aniline (1.31 mmol), 4-hydroxyphenol (1.31 mmol), 4-aminophenol (1.31 mmol), or salicylamide (1.31 mmol), as well as potassium carbonate (2.62 mmol), potassium iodide (1.31 mmol), and 10 mL of butanone, was added, and the reaction was stirred at 90 °C for 12 h. The solvent was removed under vacuum, resuspended in 100 mL of ethyl acetate, and washed with distilled water (3 × 50 mL). The organic layer was dried over sodium sulfate, and the solvent was removed under a vacuum. The crude product was purified by column chromatography (flash silica; eluent: ethyl acetate: hexane, 2:8 (*v/v*)) to provide compound (**10**). White powder; yield, 55%; ^1^H NMR (300 MHz, *DMSO_d6_*, δ ppm) δ: 7.98 (d, *J* = 8.3 Hz, 2H), 7.59 (d, *J* = 8.3 Hz, 2H), 7.30 (dd, *J* = 7.8 Hz, 2H), 7.01 (d, *J* = 7.8 Hz, 2H), 6.95 (ddd, 1H), 5.20 (s, CH_2_), 3.85 (s, CH_3_). ^13^C NMR (75 MHz, *DMSO_d6_*, δ ppm) δ: 167.2, 159.1, 139.0, 133.7, 128.3, 127.8, 126.0, 120.2, 115.5, 70.6, 51.7; column chromatography (flash silica; eluent: ethyl acetate: DCM, 3:7 (*v/v*)) to provide compound (**11**) Orange oil; yield, 82%. ^1^H NMR (300 MHz, *DMSO_d6_*, δ ppm) δ: 7.95 (d, 2H), 7.42 (d, 2H), 7.17 (t, 2H), 6.67 (t, 1H), 6.53 (d, 2H), 4.50 (s, CH_2_), 3.92 (s, CH_3_). ^13^C NMR (75 MHz, *DMSO_d6_*, δ ppm) δ: 166.4, 147.3, 144.1, 128.8, 127.7, 126.5, 127.0, 117.5, 113.0, 51.7, 45.0.; column chromatography (flash silica; eluent: ethyl acetate: hexane, 2: 8 (*v/v*)) to provide compound (12). Brown powder; yield, 50%. ^1^H NMR (300 MHz, *DMSO_d6_*, δ ppm) δ: 7.97 (d, *J* = 8.2 Hz, 2H), 7.56 (d, *J* = 8.2 Hz, 2H), 6.83 (d, *J* = 9.0 Hz, 2H), 6.67 (d, J = 9.0 Hz, 2H), 5.08 (s, 2H), 3.85 (s, CH_3_). ^13^C NMR (75 MHz, *DMSO_d6_*, δ ppm) δ: 167.3, 153.5, 152.4, 139.7, 130.2, 126.3, 126.0, 115.7, 69.8, 51.7; column chromatography (flash silica; eluent: ethyl acetate: hexane, 5:5 (*v*/*v*)) to provide compound (**13**). White powder; yield, 70%. %. ^1^H NMR (300 MHz, *DMSO_d6_*, δ ppm) δ: 7.97 (d, *J* = 8.3 Hz, 2H), 7.57 (d, *J* = 8.3 Hz, 2H), 7.35 (d, *J* = 7.4 Hz, 2H), 6.92 (d, *J* = 7.4 Hz, 2H), 4.85 (s, NH_2_) 5.13 (s, CH_2_), 3.82 (s, CH_3_). ^13^C NMR (75 MHz, *DMSO_d6_*, δ ppm) δ: 167.0, 149.7, 144.3, 140.4, 132.0, 127.7, 127.4, 116.2, 115.1, 70.2, 51.7.

#### 3.2.8. General Procedures for the Synthesis of Compounds (**15**)–(**19**)

One equivalent of compounds (**10**)–(**13**) (0.4 mmol) were added to 5 mL of THF and sodium hydroxide aqueous solution (50% *m/v*) (2 mmol), and the reaction was stirred at RT for 8 h. Then, the pH was adjusted to 4; the solvent was removed under vacuum, resuspended in 100 mL of DCM, and washed with distilled water (3 × 50 mL). The organic layer was dried over sodium sulfate, and the solvent was removed under vacuum to produce compounds (**15**–**19**). Compound (**15**)-White powder; yield, 57%. ^1^H NMR (300 MHz, *DMSO_d6_*, δ ppm) δ: 7.96 (d, *J* = 8.2 Hz, 2H), 7.56 (d, *J* = 8.2 Hz, 2H), 7.30 (dd, *J* = 7.9 Hz, 2H), 7.01 (d, *J* = 7.9 Hz, 2H), 6.95 (ddd, 1H), 5.19 (s, 2H); ^13^C NMR (75 MHz, *DMSO_d6_*, δ ppm) δ: 168.8, 158.9, 140.5, 132.4, 129.9, 129.6, 126.5, 121.6, 115.7, 70.9; Compound (**16**)-Brown powder, yield 65%. ^1^H NMR (300 MHz, *DMSO_d6_*, δ ppm) δ: 8.09 (d, 2H), 7.46 (d, 2H), 7.13 (t, 2H), 6.66 (t, 1H), 6.51 (d, 2H), 4.46 (s, 2H); ^13^C NMR (75 MHz, *DMSO_d6_*, δ ppm) δ: 168.9, 147.7, 141.9, 129.6, 129.3, 129.2, 127.4, 118.3, 114.5, 46.9. Compound (**17**)-Brown powder; yield, 40%. ^1^H NMR (300 MHz, *Acetone_d6_*, δ ppm) δ: 8.05 (d, *J* = 8.4 Hz, 2H), 7.59 (d, *J* = 8.4 Hz, 2H), 6.83 (d, *J* = 9.1 Hz, 2H), 6.67 (d, *J* = 9.1 Hz, 2H), 5.13 (s, 2H-6); ^13^C NMR (75 MHz, *Acetone_d6_*, δ ppm) δ: 168.9, 154.4, 153.3, 140.9, 133.0, 129.7, 126.6, 117.0, 116.5, 70.3. Compound (**18**)-White powder; yield, 70%. ^1^H NMR (300 MHz, *DMSO_d6_*, δ ppm) δ: 7.95 (d, *J* = 7.5 Hz, 2H), 7.54 (d, *J* = 8.3 Hz, 2H), 7.35 (d, *J* = 7.5 Hz, 2H), 6.92 (d, *J* = 8.3 Hz, 2H), 5.13 (s, 2H), 4.90 (s, NH_2_); ^13^C NMR (75 MHz, *DMSO_d6_*, δ ppm) δ: 168.9, 151.0, 143.9, 140.5, 132.4, 129.9, 126.5, 116.9, 116.0, 70.4. Compound (**19**)-White powder; yield, 21%; mp. 196.5 to 198.6 °C. ^1^H NMR (300 MHz, *Acetone_d6_*, δ ppm) δ: 8.09–8.03 (m, 3H), 7.68–7.63 (d, 3H), 7.53–7.50 (d, 2H), 4.73 (s, 2H), 5.49 (s, CONH_2_); ^13^C NMR (75 MHz, *Acetone_d6_*, δ ppm) δ: 168.9, 168.7, 157.7, 140.9, 133.5, 132.3, 131.1, 129.9, 126.6, 122.7, 122.0, 115.6, 71.5.

#### 3.2.9. General Procedure for the Synthesis of Compounds (**20**)–(**24**)

One equivalent of compounds (**15**)–(**19**) (0.88 mmol), CDI (1.32 mmol), and 5 mL of acetonitrile were stirred at RT for 1 h. Then, o-phenylenediamine (0.97 mmol) was added to the reaction and stirred for 24 h at RT. The solvent was removed under vacuum, resuspended in 100 mL of ethyl acetate, and washed with distilled water (3 × 50 mL). The organic layer was dried over sodium sulfate, and the solvent was removed under a vacuum. Compound (**20**) was purified by column chromatography (flash silica; eluent: ethyl acetate: hexane, 1:9 to 8:2 (*v/v*)) and obtained as a brown powder, with a yield of 40%; compound (**21**) was purified by column chromatography (flash silica; eluent: ethyl acetate: DCM, 2:8 (*v/v*)) and obtained as a white powder, with a yield of 25%; compound (**22**) was purified by column chromatography (flash silica; eluent: ethyl acetate: hexane, 1: 9 to 8:2 (*v/v*)); compound (**23**) was purified by column chromatography (flash silica; eluent: ethyl acetate: DCM, 2:8 (*v/v*)); compound (**24**) was purified by column chromatography (flash silica; eluent: 100% ethyl acetate). Compound (**20**)-Brown powder; yield, 40%; mp 162 to 163.7 °C. ^1^H NMR (300 MHz, *DMSO_d6_*, δ ppm) δ: 9.60 (s, NH), 8.52 (d, 2H), 8.39 (d, 1H), 8.09 (d, 1H), 8.07 (d, 2H), 7.75 (m, 3H), 7.49 (d, 2H), 7.40 (t, 1H), 7.14 (t, 1H), 5.68 (s, 2H). ^13^C NMR (75 MHz, *DMSO_d6_*, δ ppm) δ: 165.6, 158.7, 142.1, 138.9, 138.4, 129.9, 128.2, 127.9, 127.6, 126.9, 121.3, 120.1, 117.7, 115.2, 115.2, 69.1; MS/ESI *m/z* calculated for C_20_H_18_N_2_O_2_: 318.13, found: [M+H]^+^ 319.14; Compound (**21**)-White powder; yield, 25%; mp 188.3 to 189.9. ^1^H NMR (300 MHz, *DMSO_d6_*, δ ppm) δ: 9.60 (s, NH), 7.92 (d, 2H), 7.47 (d, 2H), 7.15 (d, 1H), 7.03 (d, 2H), 6.55 (d, 2H), 6.96 (t, 1H), 6.78 (d, 1H), 6.51 (t, 1H), 6.35 (t, 1H), 4.87 (s, 2H), 4.35 (s, NH_2_). ^13^C NMR (75 MHz, *DMSO_d6_*, δ ppm) δ: 166.7, 147.8, 144.3, 142.0, 135.7, 129.1, 128.9, 127.7, 127.0, 125.3, 124.8, 118.3, 118.1, 116.5, 114.2, 46.9; MS/ESI *m/z* calculated for C_20_H_19_N_3_O: 317.15, found: [M+H]^+^ 318.17. Compound (**22**)-Yellow powder; yield, 40%; mp 223.9 to 225.7 °C. ^1^H NMR (300 MHz, *DMSO_d6_*, δ ppm) δ: 9.67 (s, NH), 8.99 (s, OH), 7.98 (d, 2H), 7.54 (d, 2H), 7.15 (d, 1H), 6.95 (t, 1H), 6.83 (d, 2H), 6.77 (d, 1H), 6.67 (d, 2H), 6.60 (t, 1H), 5.08 (s, NH_2_), 4.89 (s, 2H). ^13^C NMR (75 MHz, *DMSO_d6_*, δ ppm) δ: 166.7, 154.3, 153.4, 142.1, 139.7, 138.6, 128.7, 127.7, 127.0, 125.3, 124.8, 118.1, 116.9, 116.6, 116.5, 70.8; MS/ESI *m/z* calculated for C_20_H_18_N_2_O_3_: 334.13, found: [M+H]_+_ 335.19; Compound (**23**)–White powder, yield, 35%; mp. 237 to 240 °C. ^1^H NMR (300 MHz, *DMSO_d6_*, δ ppm) δ: 9.67 (s, NH), 7.97 (d, *J* = 8.0 Hz, 2H), 7.78 (dd, *J* = 8.0 Hz,2H), 7.52 (d, *J* = 8.0 Hz, 2H), 7.17 (dd, *J* = 8.0; 7.6 Hz, 1H), 6.97 (td, *J* = 8.1; 8.0 Hz, 1H), 6.73 (d, *J* = 8.8 Hz, 2H), 6.60 (t, *J* = 8.1; 7.6 Hz, 1H), 6.50 (d, *J* = 8.8 Hz, 2H), 5.04 (s, 2H), 4.90 (s, NH_2_), 4.64 (s, NH_2_). ^13^C NMR (75 MHz, *DMSO_d6_*, δ ppm) δ: 165.1, 149.4, 143.2, 142.8, 141.4, 133.8, 127.9, 127.1, 126.7, 126.5, 123.3, 116.3, 116.1, 115.8, 114.9, 69.2; MS/ESI *m/z* calculated for C_20_H_19_N_3_O_2_: 333.14, found: [M+H]^+^ 334.17; Compound (**24**)-beige powder; yield, 10%; mp 200 to 205 °C. ^1^H NMR (300 MHz, *DMSO_d6_*, δ ppm) δ: 9.68 (s, NH), 8.02–8.00 (d, 2H), 7.79–7.78 (dd, 1H), 7.65–7.63 (m, 4H), 7.46–7.44 (t, 1H), 7.20 (d, 2H), 7.18–7.17 (d, 1H), 7.05 (t, 1H), 6.98 (t, 1H), 6.79 (d, 1H), 6.60 (t, 1H), 5.36 (s, CH_2_). ^13^C NMR (75 MHz, *DMSO_d6_*, δ ppm) δ: 165.0, 155.9, 143.2, 140.0, 134.2, 132.1, 130.6, 128.0, 127.4, 126.7, 123.7, 123.2, 121.0, 116.2, 116.0, 115.5, 113.1, 69.38; MS/ESI *m/z* calculated for C_21_H_19_N_3_O_3_: 361.14, found: [M+H]^+^ 362.16.

#### 3.2.10. General Procedures for the Synthesis of Compounds (**25**) and (**26**)

One equivalent of phthalic anhydride or 3-nitrophthalic anhydride (0.15 mmol) was treated with EDC (0.30 mmol) in 5 mL of DMF for 1 h at room temperature. Then, 4-dimethylaminopyridine (DMAP) (0.03 mmol) and compound (**23**) (0.15 mmol) were added, and the reaction was stirred for 24 h at RT. 50 mL of ethyl acetate was added to the reaction and washed with distilled water (4 × 50 mL). The organic layer was dried over sodium sulfate, and the solvent was removed under a vacuum. Compounds (**25**) and (**26**) were purified by column chromatography (flash silica; eluent: ethyl acetate: hexane, 5:5 (*v/v*). Compound (**25**)-Yellow powder; yield, 25%; ^1^H NMR (300 MHz, *DMSO_d6_*, δ ppm) δ: 9.57 (s, NH), 7.99–7.94 (m, 4H), 7.92 (d, 2H), 7.62 (d, 2H), 7.49 (d, 2H), 7.15 (m, 3H), 6.96 (t, 1H), 6.78 (d, 1H), 6.59 (t, 1H), 5.29 (s, 2H), 4.87 (s, NH_2_). ^13^C NMR (75MHz, *DMSO_d6_*, δ ppm) δ: 167.3, 166.7, 157.6, 142.0, 139.7, 138.6, 131.9, 128.7, 127.7, 126.7, 125.3, 118.1, 116.6, 115.9, 70.8; MS/ESI *m/z* calculated for C_28_H_21_N_3_O_4_: 463.49, found: [M+H]^+^ 464.52; Compound (**26**)-Yellow powder; yield, 15%; ^1^H NMR (300 MHz, *DMSO_d6_*, δ ppm) δ: 8.33 (d, 1H), 8.24 (d, 1H), 8.20 (d, 2H), 8.11 (t, 1H), 7.66 (m, 3H), 7.54 (d, 1H), 7.38 (d, 2H), 7.23 (t, 1H), 7.20 (m, 3H), 5.26 (s, 2H). ^13^C NMR (75 MHz, *DMSO_d6_*, δ ppm) δ: 165.5, 162.9, 158.2, 144.5, 143.8, 138.6, 136.4, 135.0, 129.7, 129.0, 128.3, 128.2, 127.0, 126.6, 124.2, 122.9, 122.6, 121.8, 118.9, 115.1, 111.4, 69.1; MS/ESI m/z calculated for C_28_H_20_N_4_O_6_: 508.13, found: [M+H]^+^ 509.15.

### 3.3. Enzymatic Evaluation

The enzymatic assay to evaluate HDAC inhibitory effects for all final compounds against the distinct isoforms was carried out in accordance with the previously described procedures (Lopes et al., 2021). All compounds were initially screened at 10 µM against HDAC-1 and HDAC-2. To characterize the selectivity of the most potent compounds, an assay was carried out against HDAC isoforms (HDAC 4—HDAC-11). Moreover, the IC_50_ values of the most active compounds were determined in which vorinostat (SAHA) was used as a drug control. All the compounds were dissolved in DMSO. A serial dilution of the compounds was performed in 100% DMSO with the highest concentration at 1 mM. Each intermediate compound dilution (in 100% DMSO) was diluted 10x in assay buffer to form a 10% DMSO intermediate dilution in HDAC assay buffer. 5 µL of this dilution was added to a 50 µL reaction to bring the final concentration of DMSO to 1% in all the reactions. Three independent experiments were carried out in triplicate for all experiments. The fluorescence intensity was measured at an excitation of 360 nm and emission of 460 nm using a Tecan Infinite M1000 microplate reader, and the percentage of activity was calculated using the formula: % activity = (F − Fb)**/**(Ft − Fb), where F = the intensity of fluorescence in the presence of the compound. In the absence of the test compound, the fluorescent intensity (Ft) in each dataset was defined as 100% activity, while in the absence of HDAC, the fluorescent intensity (Fb) in each dataset was set to 0% activity [45].

## 4. Conclusions

By using the chemical structure of RVT as a scaffold, we designed and synthesized new derivatives of HDAC inhibitors. Molecular docking guided structural optimizations, revealing that unsaturated linkers are not suitable for HDAC inhibition. All compounds, except (**26**), exhibited drug-like properties. All compounds were synthesized and obtained in global yields ranging from 33 to 67.3%, with inhibition values ranging from 1 to 96% for HDAC-1 and 6 to 93% for HDAC-2. Moreover, all compounds were characterized by analytical methods such as NMR ^1^H and ^13^C and mass spectrometry. The enzymatic inhibition results corroborated with the data previously obtained in the in silico study. Compound (**5**) was the most promising, with IC_50_ values of 0.44 µM and 0.37 µM against HDAC-1 and HDAC-2, respectively. We found that compound (**5**) preferably inhibited HDAC class I, mainly HDAC-1–3. All compounds were synthesized and obtained in global yields ranging from 33 to 67.3%, with inhibition values ranging from 1 to 96% for HDAC-1 and 6 to 93% for HDAC-2. Based on these results, it can be concluded that the optimization of RVT structure resulted in more potent and selective HDAC-1 and HDAC–2 inhibitors that could be used as prototypes to be applied to a variety of diseases derived from epigenetic modifications.

## Figures and Tables

**Figure 1 pharmaceuticals-15-01260-f001:**
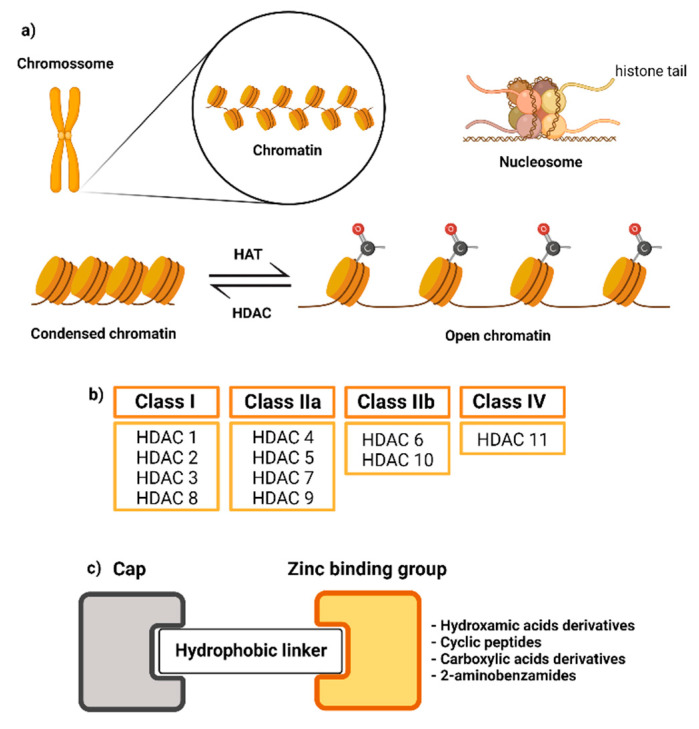
Summary of the mechanism of action of HATs and HDACs, classifications of HDACs, and the scaffold of an HDAC inhibitor (created by using BioRender.com); (**a**) Mechanism of action of HAT and HDAC; (**b**) Classification of HDACs; (**c**) Scaffold of HDAC inhibitor.

**Figure 2 pharmaceuticals-15-01260-f002:**
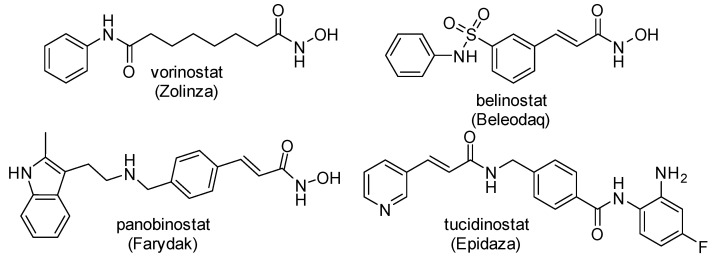
Chemical structure of licensed HDAC inhibitors.

**Figure 3 pharmaceuticals-15-01260-f003:**
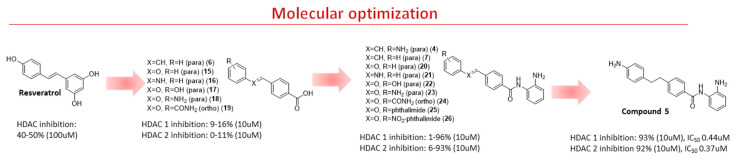
Molecular optimization to design novel RVT derivatives of HDAC-1 and HDAC–2 inhibitors.

**Figure 4 pharmaceuticals-15-01260-f004:**
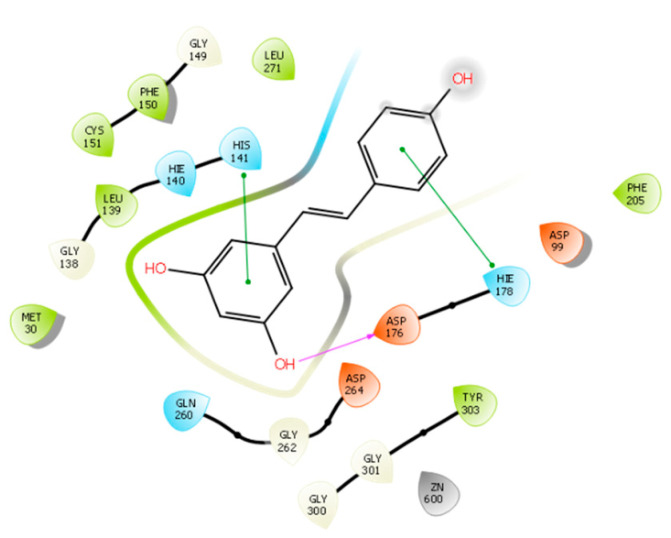
RVT interactions with HDAC-1 binding site.

**Figure 5 pharmaceuticals-15-01260-f005:**
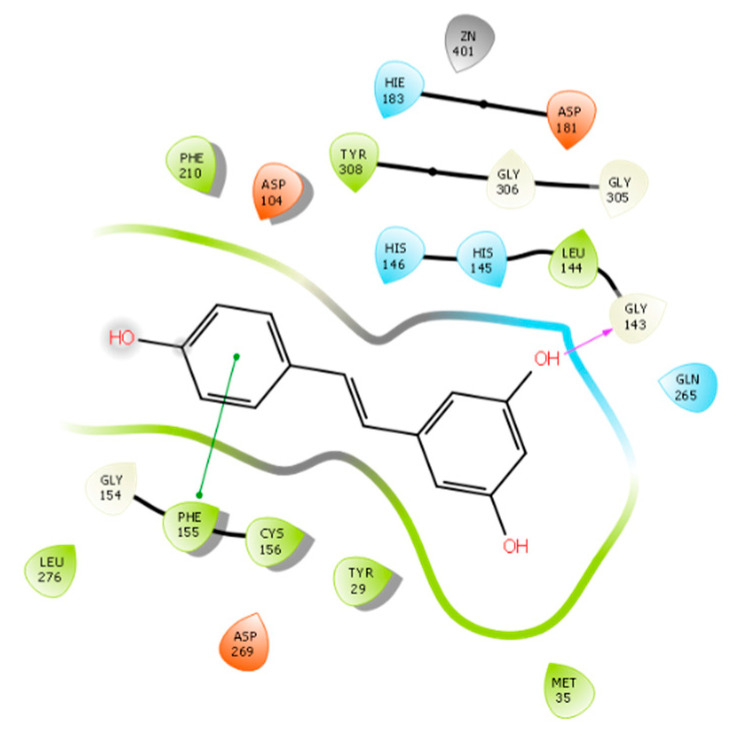
RVT interactions with HDAC-2 binding site.

**Figure 6 pharmaceuticals-15-01260-f006:**
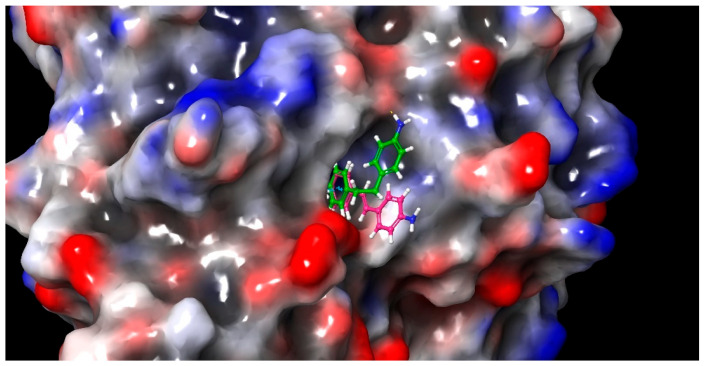
Superposition of compounds (**4**) and (**5**) in the active site of HDAC-2.

**Figure 7 pharmaceuticals-15-01260-f007:**
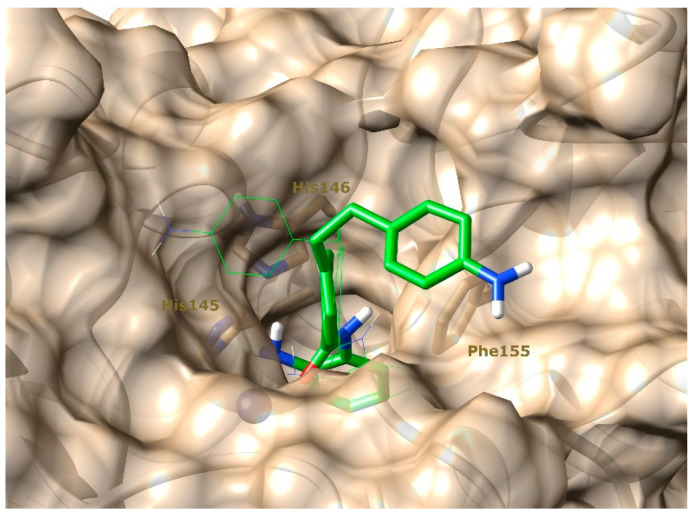
Representation of compound (**5**) conformation predicted by molecular docking (green line) and accessed during molecular dynamics simulation (green stick), in complex with HDAC-2 (tan surface and ribbons). Highlighted residues PHE155, HIS145, and HIS146, are important for the interaction with compound (**5**).

**Figure 8 pharmaceuticals-15-01260-f008:**
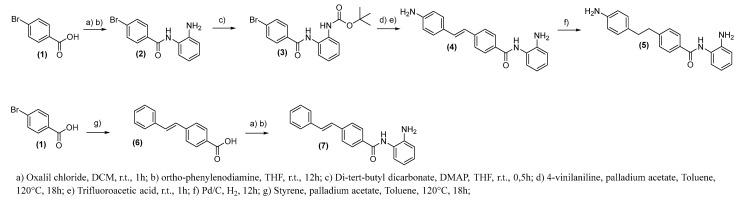
Synthesis of compounds (**4**), (**5**), and (**7**).

**Figure 9 pharmaceuticals-15-01260-f009:**
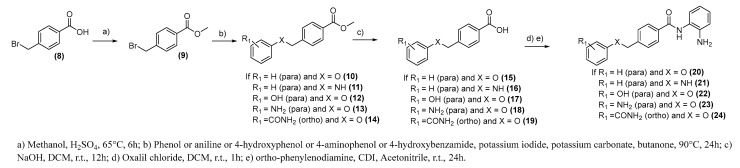
Synthesis of compounds (**20**)–(**24**).

**Figure 10 pharmaceuticals-15-01260-f010:**
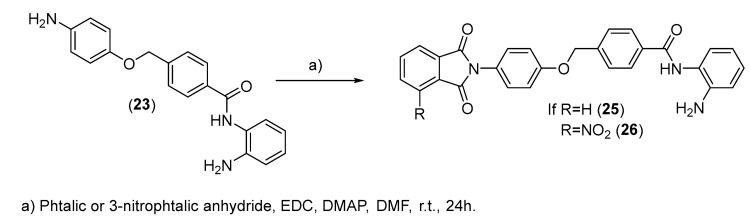
Synthesis of compounds (**25**) and (**26**).

**Table 1 pharmaceuticals-15-01260-t001:** Docking score values (Kcal/mol) for HDAC-1 and HDAC-2.

Compounds	Docking Score (DS) Values (Kcal/mol)	
	HDAC-1	HDAC-2
Resveratrol (RVT)	−5.565	−6.453
Tucidinostat	−7.606	−11.436
(**4**)	−4.462	−10.967
(**5**)	−8.724	−11.707
(**7**)	−5.331	−10.997
(**20**)	−8.542	−11.399
(**21**)	−8.760	−11.621
(**22**)	−5.068	−11.839
(**23**)	−7.268	−12.234
(**24**)	−10.763	−12.780
(**25**)	−8.764	−11.128
(**26**)	−8.437	−10.971

**Table 2 pharmaceuticals-15-01260-t002:** Initial enzymatic inhibition screening of compounds (**4**)–(**7**), (**15**), (**17**), (**19**), (**20**)–(**24**), (**25**), and (**26**), against HDAC-1 and HDAC-2.

Compound (10 µM)	HDAC-1 (%)	HDAC- 2 (%)
(**4**)	45 ± 1.2	28 ± 1.0
(**5**)	93 ± 1.0	92 ± 0.8
(**6**)	9 ± 1.0	0
(**7**)	42 ± 0.7	32 ± 0.7
(**15**)	16 ± 0.6	10 ± 0.5
(**17**)	12 ± 0.7	11 ± 0.5
(**19**)	13 ± 0.9	8 ± 0.6
(**20**)	87 ± 1.1	85 ± 1.1
(**21**)	96 ± 1.3	93 ± 1.2
(**22**)	96 ± 1.2	92 ± 0.7
(**23**)	72 ± 1.1	52 ± 0.8
(**24**)	86 ± 1.0	81 ± 1.1
(**25**)	14 ± 0.5	21 ± 0.8
(**26**)	1 ± 0.5	6 ± 0.6

**Table 3 pharmaceuticals-15-01260-t003:** Selectivity study of compound (**5**) against HDACs 3–11 at 10 µM.

Compound	3	4	5	6	7	8	9	10	11
(**5**)	82 ± 1.2	5 ± 0.4	5 ± 0.5	3 ± 0.4	15 ± 0.7	13 ± 1.0	7 ± 0.4	60 ± 1.4	15 ± 0.6

## Data Availability

Data is contained within the article and Appendix A.

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
