# Peer review of "Optimization of Resveratrol Used as a Scaffold to Design Histone Deacetylase (HDAC-1 and HDAC-2) Inhibitors"

_pharmaceuticals, 2022, doi:10.3390/ph15101260_

Round 1
Reviewer 1 Report
The manuscript is devoted to the synthesis and description of the inhibitory effect of a number of new compounds against histone diacetylases (HDAC-1 and HDAC-2).
The article is well written. I like it when the article combines molecular modeling methods with the synthesis of new compounds and biological experiments. The biological target is described, the choice of it is valid.
However, I have a number of key remarks and questions.
The binding of the RVT compound is not confirmed by crystallographic data but is probably based on the results of molecular docking. Is this true?
According to the molecular docking results, the affinity of compounds 4–26 to the binding site of HDAC-2 is higher than to the HDAC-1 binding site. Can we conclude that compounds 4–26 selectively inhibit HDAC-2? If the answer to this question is positive, then we can see contradictions by comparing the results of calculations and experiments. According to Table 2, the enzymatic inhibition of the compounds is approximately equal.
Of course, we are all well aware that the results of molecular docking may not correlate with the results of the experiment, because molecular docking is a rough estimate. Then the question arises, why present these results?
If the authors want to describe the mechanism of interaction of ligands with the binding site, it is necessary to use slightly more accurate calculation methods, for example, the induced fit docking protocol and molecular dynamic simulations.
I dare say that Glide Ligand Docking is a semi-flexible docking. Given the characteristics of the enzymes under study, it is advisable to use flexible docking tools.
I strongly advise the authors to carry out a series of molecular dynamic simulations to evaluate the behavior of the lead-compound (5) in the binding sites of both enzymes. At least for 5. Although the behavior of the RVT would also be interesting to evaluate.
You have a video card Nvidia. Try a series of calculations for 100 ns. It seems to me that the results of the molecular-dynamic simulation will greatly improve the article.
A couple of small remarks:
In pictures 4A, B and 5 A, the surface of the protein is colored according to the map of electrostatic potentials. That's the truth? If so, you should cite a legend with a minimum and maximum value.
In pictures 4C and D, 5 B and C, the labels for amino acids are difficult to read.
Author Response
Reviewer 1:
The manuscript is devoted to the synthesis and description of the inhibitory effect of a number of new compounds against histone diacetylases (HDAC-1 and HDAC-2).
The article is well written. I like it when the article combines molecular modeling methods with the synthesis of new compounds and biological experiments. The biological target is described, the choice of it is valid.
However, I have a number of key remarks and questions.
The binding of the RVT compound is not confirmed by crystallographic data but is probably based on the results of molecular docking. Is this true?
Authors: Yes. The authors used the paper from Venturelli 2013, which presented a molecular docking study of the binding mode of RVT in the active site of the HDACs, as a starting point for improving the molecule.
Venturelli, S.; Berger, A.; Böcker, A.; Busch, C.; Weiland, T.; Noor, S.; Leischner, C.; Schleicher, S.; Mayer, M.; Weiss, T. S.; Bischoff, S. C.; Lauer, U. M.; Bitzer, M. Resveratrol as a Pan-HDAC Inhibitor Alters the Acetylation Status of Jistone Proteins in Human-Derived Hepatoblastoma Cells. PLoS One 2013, 8, 1–12, doi:10.1371/journal.pone.0073097.
According to the molecular docking results, the affinity of compounds 4–26 to the binding site of HDAC-2 is higher than to the HDAC-1 binding site. Can we conclude that compounds 4–26 selectively inhibit HDAC-2? If the answer to this question is positive, then we can see contradictions by comparing the results of calculations and experiments. According to Table 2, the enzymatic inhibition of the compounds is approximately equal.
Authors: Similar inhibition values between HDAC-1 and HDAC-2 were already expected due to the high identity and similarity between them (85% identity and 93% similarity). HDAC-1 presents narrower access to the catalytic site when compared to HDAC-2, which could possibly explain a better fit of the evaluated compounds in the active site of HDAC-2 in the molecular docking studies. However, besides the docking study demonstrating a selectivity towards HDAC-2, it can not be confirmed based on the experimental data.
Of course, we are all well aware that the results of molecular docking may not correlate with the results of the experiment, because molecular docking is a rough estimate. Then the question arises, why present these results?
Authors: The authors presented the docking studies result not only to show the punctual data of the final molecules, but also the improvement achieved in the binding mode when comparing the RVT, used as the initial scaffold, and the final molecules. The docking study in our work was essential to lead us towards the structural optimization, and also demonstrated once more that the in silico approach is able to help in the drug design of new molecules.
If the authors want to describe the mechanism of interaction of ligands with the binding site, it is necessary to use slightly more accurate calculation methods, for example, the induced fit docking protocol and molecular dynamic simulations.
Authors:The authors thank the reviewer for the suggestion. The study of dynamic simulations was carried out and added in the manuscript.
I dare say that Glide Ligand Docking is a semi-flexible docking. Given the characteristics of the enzymes under study, it is advisable to use flexible docking tools.
Authors:The authors thank the reviewer for this essential comment. In order to address this issue, the authors performed a dynamic simulation study that was added to the manuscript.
I strongly advise the authors to carry out a series of molecular dynamic simulations to evaluate the behavior of the lead-compound (5) in the binding sites of both enzymes. At least for 5. Although the behavior of the RVT would also be interesting to evaluate. You have a video card Nvidia. Try a series of calculations for 100 ns. It seems to me that the results of the molecular-dynamic simulation will greatly improve the article.
Authors: The authors thank the reviewer for the suggestion. The study of dynamic simulations was varied out and added to the manuscript.
A couple of small remarks:
In pictures 4A, B and 5 A, the surface of the protein is colored according to the map of electrostatic potentials. That's the truth? If so, you should cite a legend with a minimum and maximum value.
Authors: Yes, the reviewer is correct. The authors thank the suggestion and the information about the minimum and maximum values were added to the figures.
In pictures 4C and D, 5 B and C, the labels for amino acids are difficult to read.
Authors: The authors thank the reviewer for this observation, and in order to improve the access to the information, the authors rearranged and improved the quality of the Figures in the manuscript and in the Supplementary material.
Reviewer 2 Report
Urias et al submitted manuscript, “Optimization of Resveratrol used as a Scaffold to design Histone Deacetylase (HDAC-1 and HDAC-2) Inhibitors”, is about the derivatization of resveratrol to attain good bioactivity against HDAC enzymes.
The authors did the synthesis of resveratrol derivatives and performed a screening using docking and HDAC in vitro evaluation.
Major Comments
1. There are no NMRs in the manuscript, NMR spectra are essential to confirm the molecular structure of synthesized molecules, therefore these must be added to the main manuscript to increase the synthetic section of the paper. As the author mentioned that they verified the structures using proton and carbon NMR for ALL the synthesized compounds, please add Characterised spectra and spectral data to the main manuscript.
2. No mass spectrometry (MS) data in the manuscript, which authors told in the manuscript that compounds were verified by using MS. Please add spectra of MS as supplementary information.
3. Authors focus on the molecular docking energy; therefore it is quite relevant to incorporate it into the main manuscript, "How are the molecular docking method and docking energy validated in the first place?
4. English should be correct: (a) please improve the choice of words, (b) please write it in such a way that engages the readers, say example sentence, “Disruption of this inhibition has become an interesting target for the treatment of several diseases, including cancer, hematology, neurodegenerative, immune diseases, bacterial infections, and more.”
“Disruption of this inhibition” is the subject of this sentence and should be connected to the previous sentence, which in its current form confuses the readers. As a suggestion, this could be written as, “Therefore, inhibition of HDACs has become an attractive target to enhance the chemosensitivity of cancer towards the conventional drugs, xxxx……..
Minor comments
5. Please add docking energy units.
6. Careful in writing the IC50 as it should be written as IC50 (50 must be subscript)
7. Are the synthesized compounds reported the first time? If not, please mention previously reported compounds or similar compounds with their activities, even if they weren't tested for HDACs.
8. Appropriate citations must be added to the synthetic section: The reactions are typical and therefore requires appropriate citations, so that reader could read the related articles as well.
There are several points that are not properly addressed and abrogate the quality of the manuscript to publish in the current journal.
Author Response
Reviewer 2:
Urias et al submitted manuscript, “Optimization of Resveratrol used as a Scaffold to design Histone Deacetylase (HDAC-1 and HDAC-2) Inhibitors”, is about the derivatization of resveratrol to attain good bioactivity against HDAC enzymes.
The authors did the synthesis of resveratrol derivatives and performed a screening using docking and HDAC in vitro evaluation.
Major Comments
- There are no NMRs in the manuscript, NMR spectra are essential to confirm the molecular structure of synthesized molecules, therefore these must be added to the main manuscript to increase the synthetic section of the paper. As the author mentioned that they verified the structures using proton and carbon NMR for ALL the synthesized compounds, please add Characterised spectra and spectral data to the main manuscript.
Authors: The authors apologize for this mistake in the version submitted. The data for the NMR characterization was added to the manuscript.
- No mass spectrometry (MS) data in the manuscript, which authors told in the manuscript that compounds were verified by using MS. Please add spectra of MS as supplementary information.
Authors: The authors apologize for this mistake. The data for the mass spectrometry characterization was added to the manuscript.
- Authors focus on the molecular docking energy; therefore it is quite relevant to incorporate it into the main manuscript, "How are the molecular docking method and docking energy validated in the first place?
Authors: The authors thank the reviewer for this observation, and added the validation method in the “Materials and Methodology” section as follows: “The validation of the molecular docking method was performed by redocking of the crystalized compound (20Y) and the calculation of the Root Mean Deviation Square (RMSD)”.
- English should be correct: (a) please improve the choice of words, (b) please write it in such a way that engages the readers, say example sentence, “Disruption of this inhibition has become an interesting target for the treatment of several diseases, including cancer, hematology, neurodegenerative, immune diseases, bacterial infections, and more.”
“Disruption of this inhibition” is the subject of this sentence and should be connected to the previous sentence, which in its current form confuses the readers. As a suggestion, this could be written as, “Therefore, inhibition of HDACs has become an attractive target to enhance the chemosensitivity of cancer towards the conventional drugs.
Authors: The authors thank the reviewer for this comment, and the suggestion was added to the manuscript.
Minor comments
- Please add docking energy units.
Authors: The energy unit of the docking study was added.
- Careful in writing the IC50 as it should be written as IC50 (50 must be subscript).
Authors: The authors thank the reviewer for this observation, and all corrections have been made.
- Are the synthesized compounds reported the first time? If not, please mention previously reported compounds or similar compounds with their activities, even if they weren't tested for HDACs.
Authors: Two of the described compounds have been published before, and the citations were added to the manuscript with their inhibitory activities. The other molecules are new.
- Appropriate citations must be added to the synthetic section: The reactions are typical and therefore requires appropriate citations, so that reader could read the related articles as well.
Authors: The authors thank the reviewer for the suggestion. The proper citations were added to the methodology section.
Reviewer 3 Report
In this manuscript, Urias and co-workers try to build new molecules using RVT as the scaffold to develop HDAC inhibitors.
1. The biggest problem is that the authors have not reported any statistical errors in most results!
2. From Table 2, the most potent inhibitors seem to be 21 and 22. And they are even slightly better than compound 5. The authors should perform the IC50 assay and HDACs selectivity assay on 21 and 22 too.
3. The authors should correlate and compare the molecular docking and the inhibition screening results and discuss the trends. And also show the docking results of other molecules as supplementary figures.
4. The authors should show the raw IC50 curves plotting.
5. Figure 4 and Figure 5, are unclear, the authors should label which is which. And what PDB models are used.
6. Since most of the molecules are with similar structural subunits, the authors should expand the discussion on what makes 5, 21, 22 better than the others and why.
7. In the “Conclusion” section, two sentences at line 431 and at line 439 are identical.
Author Response
Reviewer 3:
In this manuscript, Urias and co-workers try to build new molecules using RVT as the scaffold to develop HDAC inhibitors.
- The biggest problem is that the authors have not reported any statistical errors in most results!
Authors: The authors thank the reviewer for this important comment. We corrected it by including in the manuscript.
- From Table 2, the most potent inhibitors seem to be 21 and 22. And they are even slightly better than compound 5. The authors should perform the IC50 assay and HDACs selectivity assay on 21 and 22 too.
Authors: The authors thank the reviewer for the suggestion. This observation is pertinent however, the reason to continue evaluating compound 5 was physico-chemical properties, such as water solubility. Compound 5 (arylamine) has better water-solubility for the assays compared to 21 and 22.
- The authors should correlate and compare the molecular docking and the inhibition screening results and discuss the trends. And also show the docking results of other molecules as supplementary figures.
Authors: The authors thank the reviewer for this suggestion. The correlation between the molecular docking study and the enzymatic evaluation was added to the manuscript, and the docking results for the other molecules were added to the supplementary material.
- The authors should show the raw IC50 curves plotting.
Authors: The authors thank the reviewer for the suggestion, the IC50 curves of compound (5) were added to the Suplemmentary material (Figures S1 and S2).
- Figure 4 and Figure 5, are unclear, the authors should label which is which. And what PDB models are used.
Authors: The authors thank the reviewer for the comment. The Figures were rearranged in the manuscript and in the supplementary material, and the label was added to the figures.
- Since most of the molecules are with similar structural subunits, the authors should expand the discussion on what makes 5, 21, 22 better than the others and why.
- In the “Conclusion” section, two sentences at line 431 and at line 439 are identical.
Authors: The authors thank the reviewer for the comment. One of the lines was excluded.
Round 2
Reviewer 2 Report
The authors revised the manuscript accordingly and significantly improved from the previous version. However, NMR spectra of compounds are still missing from supplementary information.
Author Response
NMRs spectra were now added to the supplementary material. Thanks for your comments.
Reviewer 3 Report
I don't see the authors' response on the point 6. I think it is good to have that in discussion.
Author Response
The authors thank the reviewer for this reminder. Based in that, we included in the last paragraph of discussion the following statement:
"Although compounds (5), (21) and (22) were the most active inhibitors for HDAC-1 and HDAC-2, we selected compound (5) for the IC50 and selectivity studies. The reason for that was due to favorable pharmacodynamic and ADME profiles. It is well established in the literature the importance of the hydrophobic character of the linker. As described [16], the 11Å channel in the active site of HDAC-1 and HDAC-2 presents hydrophobic amino acid residues, which was taken into account in the decision of compound (5) compared to (21) and (22). The presence of arilamine In addition, the intermediate clogP values (logP < 3) and good water solubility for (5) motivated the continuity of the studies with this compound. We found fo compound (5) sub micromolar IC50 values against HDAC-1 and HDAC-2, and an interesting selectivity towards HDAC Class I."
Moreover, the have found for (5) an additional hydrogen bound interaction involving N-H (amino (arilamine subunit) and the aminoacid Leu276 (supplementary material).